# Territorial Fragmentation and Renewable Energy Source Plants: Which Relationship?

**Lucia Saganeiti \*, Angela Pilogallo, Giuseppe Faruolo, Francesco Scorza and Beniamino Murgante**

School of Engineering, University of Basilicata, 85100 Potenza, Italy; angela.pilogallo@unibas.it (A.P.); giuseppe.far88@gmail.com (G.F.); francesco.scorza@unibas.it (F.S.); beniamino.murgante@unibas.it (B.M.)

**\*** Correspondence: lucia.saganeiti@unibas.it; Tel.: +39-340-968-4175

**Abstract:** Renewable Energy Sources (RES) are part of the solution to tackle the global problems of climate change and carbon emissions. Programs and policies at different levels are continuing to promote new RES farms, posing a relevant challenge to regional planners and administrators: how to manage landscape transformation and territorial fragmentation to find a really effective sustainable arrangement for these kinds of technologies? Most effects induced by RES (land-use change, land take, diminishing aesthetic values, loss of habitat quality), without a doubt, depend on the location and the spatial pattern of the plants, the relative distance between them, the extension of secondary infrastructures and their technical characteristics. This work takes part in the debate, originating from the need to establish a monitoring system for this kind of new territorial transformation and discusses the implementation of a sprinkling fragmentation index (SPX) in order to assess the current regional settlement structure of RES farms. Our case study concerns the Basilicata region (in Southern Italy), a very low-density area which over the last decade has undergone a relevant increase in the installation of RES technologies, not supported by an effective planning framework. The evolution of the regional energy system has been strongly influenced both by incentive policies and by (weak) urban and territorial planning policies. This approach could be a valuable contribution both in identifying a fragmentation threshold beyond which the expected negative impacts outweigh the benefits, and in providing a useful procedure for the management of future installations.

**Keywords:** RES; fragmentation; sprinkling; Basilicata region; low density

## 1. Introduction

Following global challenges on climate change and the relevant framework of international agreements on CO2 emission reduction, a widespread local policy-making strategy was promoted at different scales. As a matter of fact, Renewable Energy Sources (RES) represent a relevant component of the solution toolbox adopted by public and private operators in order to tackle global concerns on climate change, driving territorial development towards a low carbon economy and sustainability principles [1].

If we focus on RES plants intended as a new component of territorial settlements and, therefore, compare their subsequent anthropic pressure on the metrics already adopted to measure urban growth, we clearly realize how RES development has to be considered a critical concern for current urban and territorial planning.

Although the spread of RES technologies is strongly characterized by specific installation conditions, widespread territorial impacts had been generated on sensible components such as: landscape, rural areas, natural sites, and cultural heritage [2–4].

RES-related impacts already analyzed in recent scientific literature [5–9] are: change in land use, land take, natural habitat fragmentation, aesthetic impacts and micro-climate alteration.

RES development is growing rapidly [10] and such a condition is one of the main causes of the lack of integration between energy planning (that in the Italian experience has been promoted without a clear analysis of the spatial dimension of the phenomena) and the urban and territorial planning system, traditionally unsuitable to be adapted in the short run to include arising instances that derive from new territorial transformation trends [11,12].

At the moment, this type of transformation is regulated in a fragmented and sectorial way, with consequences that at a local level, risk being completely neglected and obscured by the global need to tackle climate change by reducing greenhouse gas emissions.

Such an issue highlights the need for an integrated territorial monitoring system allowing decision makers to provide effective policy making and governance of a territorial transformation able to demonstrate the sustainability of the results from the dual perspective of both global needs and preservation of local values.

This work fits into the debate by testing the sprinkling index (SPX) [13], which has already been successful in representing the territorial fragmentation thanks to a disorganized growth of the settlement system [14–16], to describe the manner and intensity (increasing fragmentation degree) with which the spread of RES installations has ended up modifying (or even compromising) the quality of both the landscape and the habitats. Landscape fragmentation is generally defined as any kind of process that divides habitats, ecosystems or flora and fauna populations into smaller and more isolated units (fragments) [17]. Specifically, territorial fragmentation, refers to morphological changes in urban areas and their disorganized dispersion in the space [18]. Generally, the main factors that cause territorial fragmentation are infrastructure construction, urban growth and dispersion of rural settlements [13,19,20]. Considering RES plants as a new component of territorial settlement, we assume that their installation becomes an additional cause of the territorial fragmentation process. In reality, over the last decade, programs and policies at different levels have continued to promote new RES installations, and in some contexts, such as in the Basilicata region, this has come about in an uncontrolled manner generating a further fragmentation of the territory. It is important to analyze the process of territorial fragmentation seeing that the quality of public services in the urban system, in the management of economic and environmental sources, the productive efficiency and the quality of the ecosystems, are highly influenced by the spatial disposition of the individual elements that constitute the urban settlement [21]. It is useful, therefore, to investigate the fragmentation caused by the RES, at regional level, using the SPX index.

The case study is the Basilicata region, in Southern Italy, where the boost given by energy policies (Regional Environmental Energy Plan—PIEAR, approved in 2010) to the development of RES installations has not been matched by any regional planning regulatory framework. Together with the presence of intrinsic territorial characteristics that are advantageous for the production of electricity from renewable sources (wind, presence of unforested hills, presence of agricultural areas not very profitable or not cultivated with certified crops) this has led to a quantity of turbines that, according to the Energy Services Provider, at the end of 2017 accounted for just over 25% of the national wind farms.

The methodological approach of this research is oriented towards adopting an indicator of territorial fragmentation previously used to analyze urban development [13,22] for the assessment of the impact produced by RES in increasing sprawl [23–25] and sprinkling [26,27].

Based on the extent and distance between RES installation aggregates, the SPX index has been calculated for three periods (before 2008, from 2008 to 2013, from 2013 to 2017) by dividing the regional area into a 1 km$^2$ grid.
The results obtained, considering the whole settlement system (building stocks and RES farms), indicate that in the low-density context of the Basilicata region, the effect of RES has produced a significant increase in territorial fragmentation.

This study provides additional motives to consider the RES spread as not fully sustainable process. In fact, if only the components of $CO_2$ reduction, energy saving for the users, and generalized

benefits for the energy market are considered, the RES system could be intended as an advantageous process. If local territorial values come into the evaluation models, instead, the results may be different and even opposite. In this perspective, it is interesting to study the fragmentation process as it can help to provide indicators of land use in the support of sustainability principles (environmental, social, and economical).

## 2. Study Area

The study area is the Basilicata Region, located in Southern Italy and extending about 10,073 km². According to the Italian National Statistical Office (ISTAT, [28]), the resident population amounts to about 567,000 inhabitants [28], distributed between the two provinces of Potenza (368,251 inhabitants) and Matera (198,867 inhabitants) and a total number of 131 municipalities.

Since the 1950s great transformation dynamics have occurred, driven both by the demographic increase and the subsequent need for housing, and the general economic growth that has led to the establishment of industrial areas and production centers. Since the early 2000s, with the development of technology for the exploitation of renewable resources for energy production, a relevant increase in RES farms has come about. According to the national Energy Services Provider (GSE [29]), the Basilicata region is currently the Italian region with the highest percentage of wind farms on the national territory (25%), followed by the Apulia region (21%).

In order to fully understand the intensity and the means of spreading of transformation dynamics associated with RES plants, it is useful to compare the percentage of power produced in the Basilicata region in relation to the national context. In reality, as highlighted in Figure 1, whereas the percentage of the number of installations and the power generated for the other regions are similar, in this study area this difference is relevant.

It can be noted, moreover, that throughout most of the Italian territory the power produced is greater than the number of turbines whereas in the Basilicata and Tuscany regions, the opposite occurs.

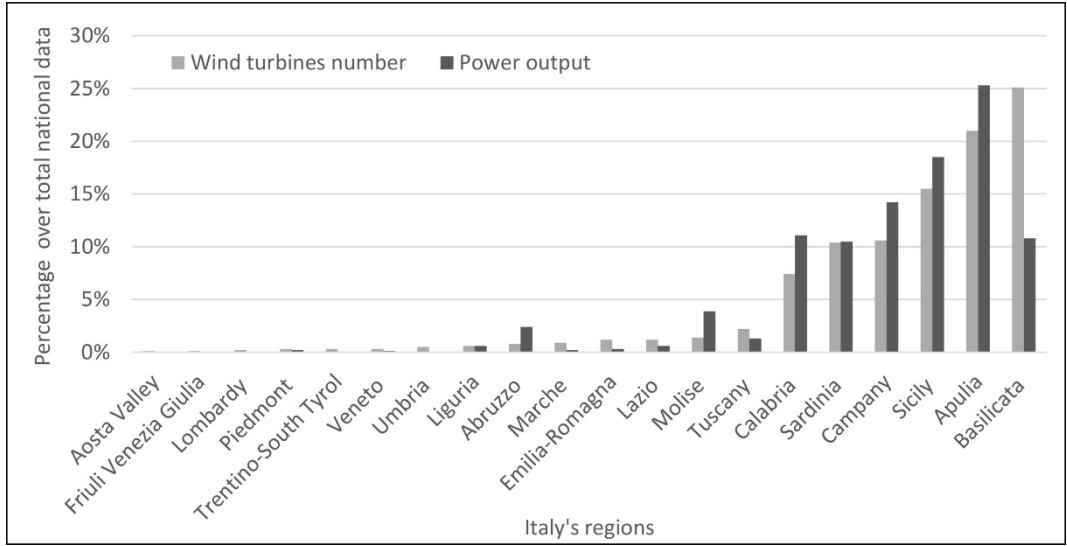

**Figure 1.** Distribution of wind turbines per number and power output (in percentage) for each region of the Italian territory. Chart elaborated on Energy Service Provider data [29].

The Figure 2 shows the geographical layout of the Italian regions. The disparity in the distribution between northern and southern Italy is evident both in terms of power output and the number of wind turbines installed. The southern regions (Campania, Basilicata, Apulia, Calabria, and Sicily) are those with the highest percentage classes.

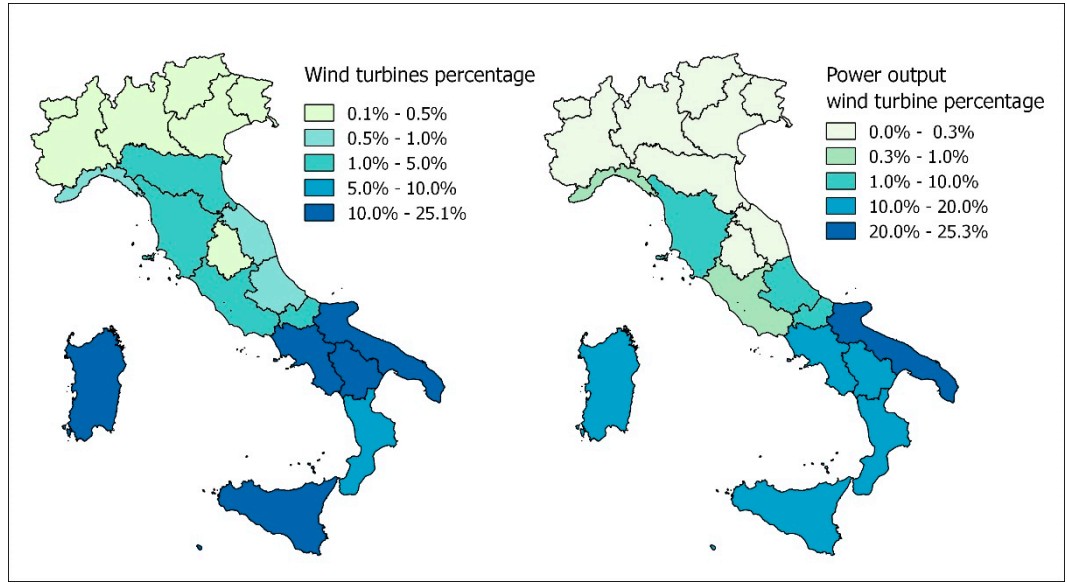

**Figure 2.** Number of farms on the left and power output percentages on the right [29].

It is significant that the boost that technologies for energy production from renewable sources have received during the last decades, has not produced uniform effects throughout the Italian territory. The explanation lies in the policies, both financial and administrative, that the Basilicata Region has implemented in order to increase the percentage of energy produced from renewable sources. Financing and economic incentives were in fact provided by the Basilicata Region which, in the frames of the European Regional Development Fund (ERDF) and the Rural Development Program (EAFRD), has financed RES farm installations supporting different kinds of beneficiaries (private citizens, small and medium enterprises, local authorities, and administrations) and, therefore, different types of wind turbines.

From an administrative point of view, the instrument used to promote RES development is streamlining of the authorization procedure. According to the PIEAR, projects involving turbines with a power output lower than 1 MW and a number of wind generators that do not exceed 5, do not follow the procedure established on a national level but can only be authorized after a Certified Notice of Commencement of Construction Works (SCIA) has been issued. This is a statement that allows private operators to start, modify or stop production activities (craft, commercial, industrial), without any lead-time related to preliminary checks and inspections by the competent authorities. Additional procedures and documents are required for farms with a power output that exceeds 200 kW and for farms close to Natura 2000 sites.

Figure 3 shows that most of the turbines that make up the regional wind farm have consistently been small sized. This trend has further increased in the last decade.

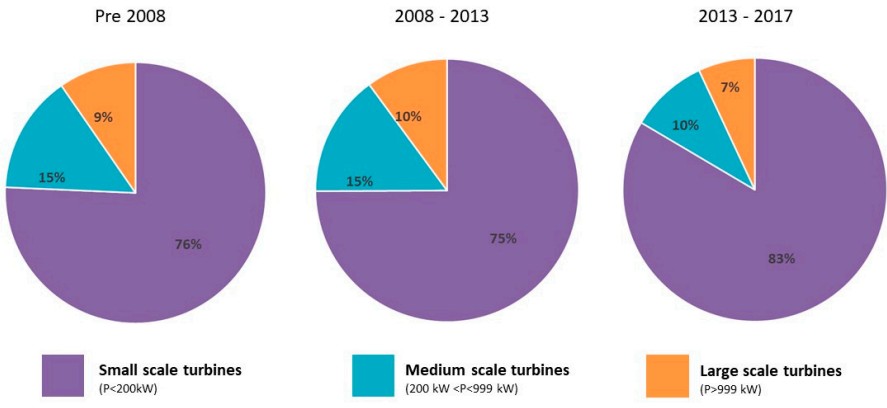

**Figure 3.** Regional composition of wind farm over the three time periods.

In view of this relevant impulse towards territorial transformations connected to the installation of new wind farms, no monitoring system has been implemented by the regional government.

## 3. Methodology

The aim of this work is to assess the fragmentation that results from territorial transformation processes in the low-density context of the Basilicata region. This study area, although characterized by a significant and well-established depopulation phenomenon, has undergone significant land changes in recent decades due to different dynamics.

In the post-WW II period, the real estate assets increased, driven by a relevant demographic growth. The subsequent construction of new buildings was instead mainly attributable to the need for residents to move into new, more functional and equipped aggregates that better responded to the new requirements. In some cases, these dynamics were further intensified by re-locations planned by the administrations and due to hydrogeological instability phenomena.

On the other hand, in the last decade, the main driving force for land changes has been energy production by RES.

The aim of this paper is to evaluate the effectiveness of the SPX index as a provider of territorial fragmentation no longer due to the evolution of the building settlement system, but also to the growth in the number of widespread RES installations. For this reason, the methodology is based on the comparison between the results achieved considering only the building stocks and what has been attained considering both buildings and RES systems. Figure 4 shows the diagram of the methodology adopted in this paper.

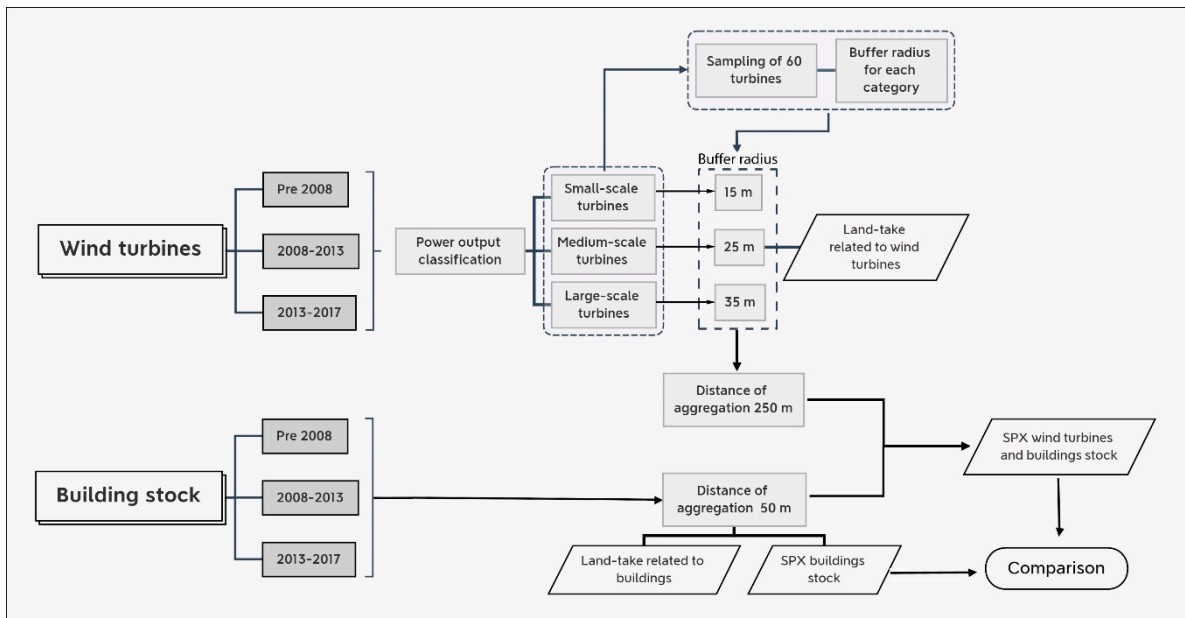

**Figure 4.** Conceptual map of methodology.

Prior to the fragmentation assessment, a significant preliminary task concerning the collection of the RES farms database was carried out (this was not available on the regional geoportal nor in document form at the municipal technical offices).

Thus, the following paragraphs describe the three main phases of the work: dataset constructing; formation of aggregates and sprinkling index (SPX) assessment.

### 3.1. Dataset Constructing

For the purposes of processing and reconstructing changes in the settlement system of the Basilicata region, it was necessary to integrate different data sources. At this stage, the lack of official spatial data made the task very difficult.

RES farm datasets were compiled using at first data from the GSE website. The vectorial file thus obtained was subsequently edited and integrated through photo-interpretation and by integrating the data available on the regional geoportal (RSDI) limited to large-scale turbines only. By comparison with orthophotos at different time intervals, the RES dataset was divided into three time periods: (i) RES existing before 2008, (ii) RES existing in 2013 and installed between 2008 and 2013 and (iii) RES existing in 2017 and installed between 2013 and 2017. In 2006 the first wind turbine was installed in the study area so it was detectable only in 2008, the year of the first useful orthophoto available. The year 2017, instead, corresponds to the date of the last available orthophoto for the entire regional territory.

Furthermore, according to the PIEAR classification, wind turbine data were grouped into three classes highlighted in Table 1.

**Table 1.** Increase in number of wind turbines for each class and for each period.

| Wind Turbines | Power Output (kW) | Number | | | Total per Class |
|---|---|---|---|---|---|
| | | Pre 2008 | 2008–2013 | 2013–2017 | |
| Small-scale turbines | <200 Kw | 165 | 474 | 1129 | 1768 |
| Medium-scale turbines | 200Kw ≤ Power < 1000Kw | 32 | 96 | 74 | 202 |
| Large-scale turbines | ≥1000 Kw | 21 | 65 | 61 | 147 |
| **Total per Year** | | 218 | 635 | 1264 | 2117 |

RES installation results in a higher land take than the area corresponding to the sum of individual grasslands. For this reason, in order to integrate an area including the land taken by access roads and auxiliary infrastructures into the computing, a method based on the hypothesis that cumulative land take is proportional to the power output of farms, was used. The hypothesis was verified by a sample of sixty wind turbines (twenty per class). For each one, the footprint area perimeter was digitized and correlated to the power output. The graph in Figure 5 shows a positive correlation, with $R^2$ of 0.79, between the area occupied by the sampled wind turbines and their power output. The three clusters of the wind classes considered, were also identified.

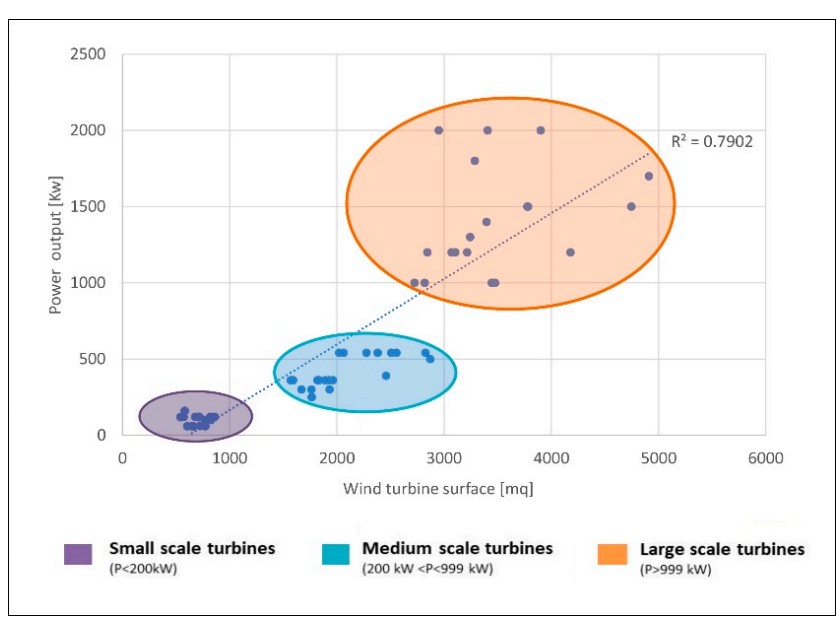

**Figure 5.** Correlation between the wind turbine surface and the power output.

On the basis of these samples, in order to evaluate the land-take related to the installation of wind turbines on the entire regional territory, it was assumed that the area occupied by each turbine is equal to the surface of a circle with buffer radius proportional to the power output class.

Subsequently, the following formula was applied in order to calculate the buffer radius for each class of wind turbines:

$$Buffer\ radius = \sqrt[2]{\frac{\sum_1^n taken\ up\ area}{n° turbines} \times \frac{1}{\pi}}$$

Where, for each turbine class: $\sum_1^n taken\ up\ area$ is the sum of the digitized surfaces (in square meters) including the pitches and the access roads to the individual turbines; $n° turbines$ corresponds to the number of wind turbines sampled, i.e., 20.

Table 2 summarizes the values thus calculated and then used to assess current land take related to wind farms.

**Table 2.** Buffer radius calculated for the three classes of the turbines taken into consideration.

| Turbines | Buffer Radius (m) |
|---|---|
| Power output < 200 Kw | 15 |
| 200 Kw ≤ Power output < 1000 Kw | 25 |
| Power output ≥ 1000 Kw | 35 |

The graph in Figure 6 shows the temporal evolution divided by the three classes. In particular, the histogram represents the number of wind turbines per class and for each period analyzed, whereas the line diagram shows the area occupied (in hectares), calculated with the buffer radius.

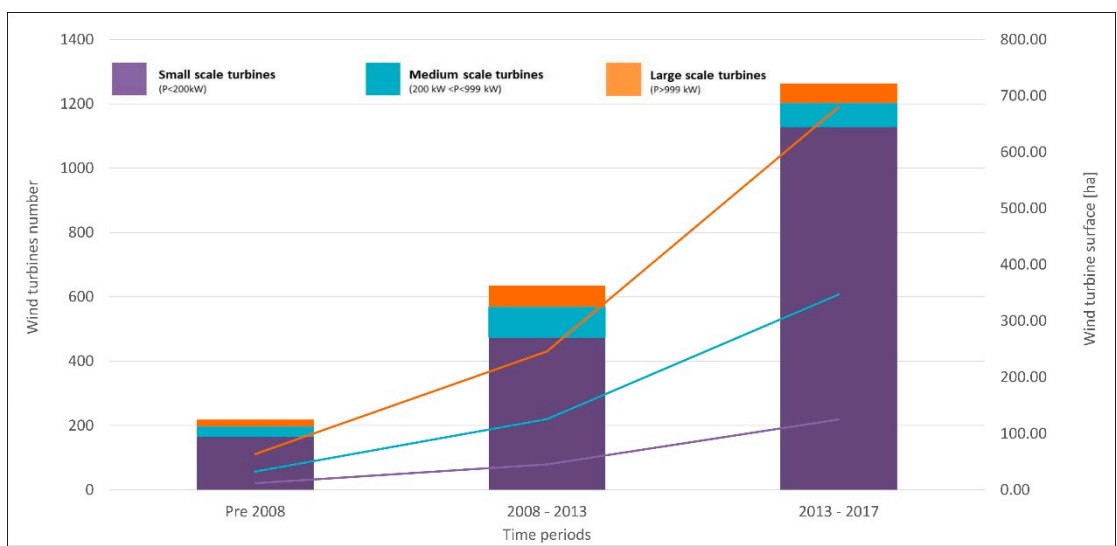

**Figure 6.** Graph of the number of wind turbines (histogram) and area occupied in hectares (lines) for each period and for each class.

The classification of wind turbines according to power output provides a clear picture of the current status of the study area in the three periods. It is clear that although the number of small-scale turbines is very high, the largest area (in terms of occupied surface) is that of the large-scale class (680 ha). In terms of the spatial distribution however, the small-scale turbines, due to their high number, are scattered over the territory in a fragmented manner.

Concerning the building stocks, the dataset was collected on the basis of the regional technical map [30] and dataset deriving from previous works [14,22] carried out on the same study area and considering the same time intervals of the RES.

### 3.2. Aggregates

Since the effects of the RES installations (e.g., loss of habitat quality) extend over a surface larger than the sum of the calculated areas, wind turbines have been aggregated with a maximum distance of 250 m between them [31]. It is also assumed that the areas between turbines partially lose their suitability for land use previous to the plant's construction. As a matter of fact, the area of the aggregate includes not only the grassland, the access road and the technical rooms of the plant itself, but also a measure of the total area irreversibly taken from its natural vocation. It should be specified



that an aggregate is considered to be a polygon containing two or more polygons whose distance (between the polygons of the buffer radius for RES and between the individual polygons for buildings) is less than a predetermined threshold.

This aggregation allows to calculate, in the case of RES, the amount of surface occupied after the installation of a single turbine. This is mainly due to the proximity/aggregation of RES installations to each other rather than the direct presence of a single wind turbine. Small-scale turbines placed at short distances from each other will be aggregated, while large-scale turbines placed at greater distances will be considered individually.

In the case of the building stocks, the single buildings were aggregated considering the threshold of 50 m as in previous works [22]. As far as the building stocks are concerned, the construction of the aggregate allows to consider not only the polygon of the building but also the land take related to roads, parking areas and all public services related to the construction. In this perspective, a building aggregation can be considered as a new method to represent groups of buildings within an urban area. The idea is different, therefore, from the traditional concept of an urban center which includes the continuity of buildings and intercluded lots.

The construction of the aggregates makes it possible to shift the analysis from punctual to spatial, which in turn enables to study the spatial arrangement of the elements and the spatial density. This phase allowed us to understand how transformations occurred: an increase in the number of elements does not always correspond to a higher number of aggregates; therefore, two or more elements can be considered an aggregate when distances between each other are lower than the pre-established threshold. As the number of aggregates increases, so does the fragmentation. On the other hand, a decrease in the number of aggregates corresponds to a densification due to the new buildings and /or RES installations set up between existing elements to form a single and larger aggregate.

### 3.3. Sprinkling Index (SPX)

Territorial fragmentation was assessed using the spatial sprinkling index (SPX) [13,32] by dividing the study area in a 1 km$^2$ side cell grid. The SPX is based on the Euclidean distance between two or more geometries within each cell assuming that the most compact agglomeration growth form is the circular one. In this case the geometries are made up of the RES aggregates and those of the building stocks. It is expressed by the following formula:

$$SPX = \frac{\sum \sqrt{(x_i - x^*)^2 + (y_i - y^*)^2}}{R}$$

where $x_i$ and $y_i$ represent the position of the centroid of each aggregate while $x^*$ and $y^*$ are the centroid coordinates of the largest aggregate present in each mesh of the grid. $R$ is the radius of the equivalent circular area that corresponds to the sum of the areas of all the aggregates.

The SPX ranges from 0 to infinite where the zero value represents "non-fragmented" cells, i.e. a single aggregate with an area considerably lower than the grid or an aggregate with area of exactly 1 km$^2$. For this reason, it is always advisable to correlate the SPX index spatial distribution with the total footprint for each cell. Null SPX values correspond to "non-urbanized", i.e., cells with no aggregate.

According to the classes summarized in Table 3, seven territorial fragmentation categories were defined. The degree of fragmentation of a territory grows with the increase of the SPX index.

**Table 3.** Fragmentation degree according to sprinkling index (SPX) index.

| Fragmentation Degree | SPX |
| --- | --- |
| Not urbanized | Null |
| Not fragmented | SPX = 0 |
| Low fragmentation | 0 < SPX < 25 |
| Medium-low fragmentation | 25 ≤ SPX < 50 |
| Medium fragmentation | 50 ≤ SPX < 100 |
| Medium-high fragmentation | 100 ≤ SPX < 200 |
| High fragmentation | 200 ≤ SPX > 500 |

The SPX index is considered a useful tool to monitor the fragmentation linked to the different components of the settlement system. It allows to simultaneously consider the extension, the shape and the distance between the different geometries present in the same cell over different time periods.

## 4. Results and Discussion

The SPX index was calculated on a regional scale in three different periods considering a 1 square km grid. Building stocks and RES farms have been considered simultaneously.

Figure 7 shows the map for the last time period (2013–2017) in which the main type of transformation corresponds to a medium fragmentation (23.64% of the total amount). As can be seen, the only areas that have remained unchanged and not affected by anthropization are located in the southern part of the region (areas included in the Pollino National Park), in the western part (the Agri Valley, where the oil industry was established in the 90s).

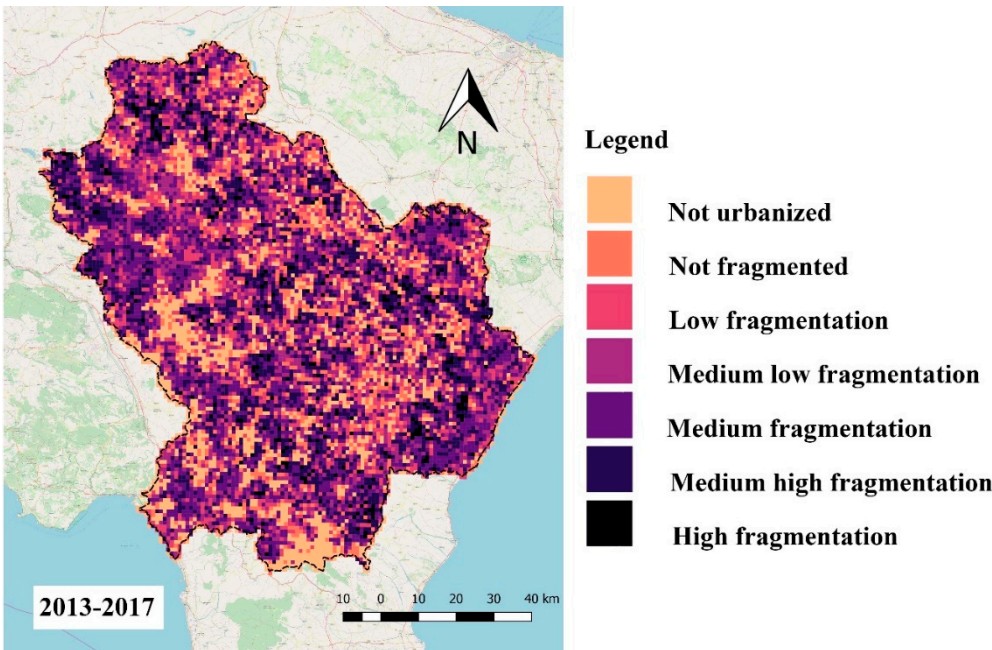

**Figure 7.** Spatial distribution of the SPX fragmentation index in the last time period (2013–2017).

Considering the changes in the spatial distribution of the fragmentation degree between 2008 and 2017, and in looking at the entire settlement system consisting of both building and RES plants, a contraction of 'non-urbanized' and 'non-fragmented' classes mainly in favor of 'low' and 'medium-low' fragmentation classes can be observed.

In observing the entire period, the variation trends of most of the fragmentation classes, appear almost stationary. The exception is the 'medium' and 'medium-high' fragmentation classes, which increase significantly over the period 2008–2014 and then decrease in 2018. This is significant of the ways in which the development of RES technologies has led to territorial transformation. In fact, in the first time period, the most advantageous areas have been "colonized" with large-scale turbine installations. Having (rapidly) reached the production limit set by the PIEAR for this type of installation, wind farms were densely populated with smaller turbines (small-scale class).

Figure 8 shows the distribution of the fragmentation values as at the last available date, i.e., 2017.

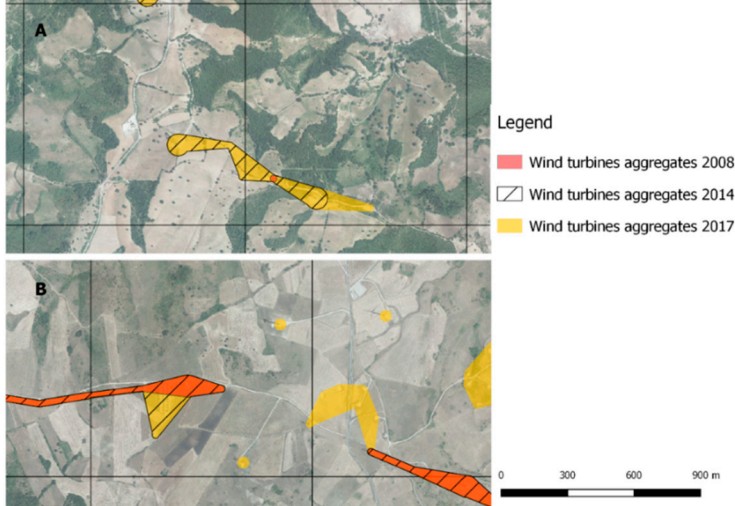

**Figure 8.** Comparison between areas where compaction (**A**) or fragmentation (**B**) occurred.

These values can essentially be traced back to two different types of dynamics. Over the entire period, in fact, new installations have expanded existing wind farms, thus increasing the surface of the same aggregate (Figure 8A). Alternatively, the new turbines were installed in previously unmanned areas or at a distance of more than 250 m from the previous installations, thus leading to the emergence of new aggregates and an increased fragmentation (Figure 8B).

In order to interpret the transformation dynamics, the values of the SPX index have been reported for each of the three years considered, on a graph representing the areas classified in the aforementioned seven categories.

The role of RES development on territorial fragmentation emerges even more clearly when comparing the same graph drawn up considering only building stocks and the whole settlement system (building stocks and RES farms).

Figure 9 shows the surfaces in hundreds of hectares, falling into each fragmentation class considering only the 'building stock component and both components together ('buildings stock and 'RES farms'). The aim is to highlight the increase in the areas classified as 'high fragmentation' considering the contribution of RES farms. The class of "non-urbanized" areas was reduced, (due to RES farms) by 41 hectares in the last time period.

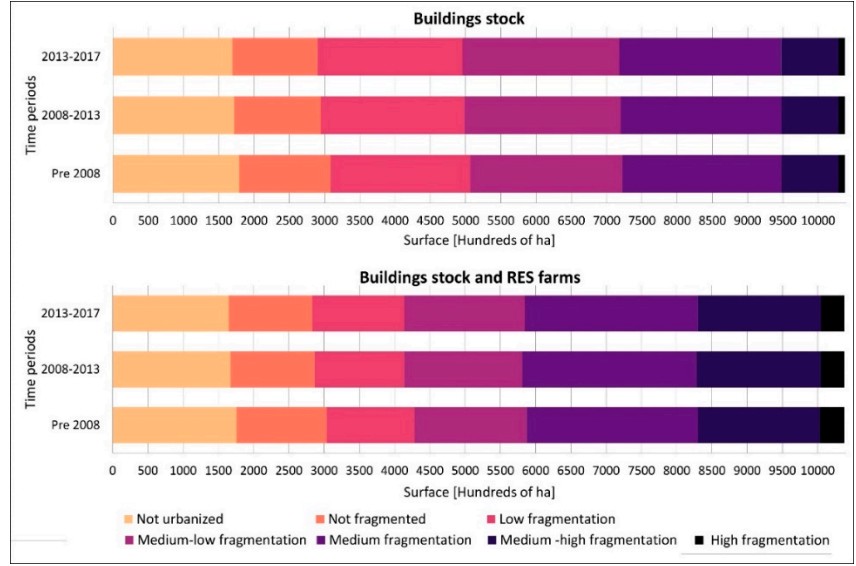

**Figure 9.** Fragmentation degree of building stocks and Renewable Energy Systems (RES) farms plus building stocks.

Compared to the fragmentation caused by building stocks only, RES farms have caused a shift from 'non-fragmented' to 'medium-low' for a total of 13 hectares in 2008, 14 hectares in 2014 and 12,.8 hectares in 2018. On the other hand, an equal amount of area changed from the 'medium' to the 'high fragmentation' class.

As can be seen, there is a marked difference in the areas classified with the highest degree of fragmentation: considering only the building stocks, these are just over 80 square kilometers; considering also the RES these areas increased to over 330 square kilometers during the last time-period.

In order to highlight that the process of territorial fragmentation, which came about in recent years, is more attributable to the installation of new wind turbines than to the construction of new buildings, the new areas occupied by the two components of territorial settlements ('Building stocks' and 'RES farms') analyzed in the three time phases were compared.

In observing Figure 10, it can be seen that the trend has completely inverted. The first interval refers to the surface of building stocks and wind farms between 2000 and 2008. The new area occupied by the building stocks (92%) is significantly larger than that of the wind farms (8%). In the second period (2008–2013), the surface of wind farms began to grow significantly, affecting 23% of the territorial transformation. The progressive decrease of territorial transformations due to building stocks leads to a total inversion of the trend in the last period. In the third period, in fact, the surface modified by new wind farms is more than twice as large as the area sealed due to new construction. The installation of new wind turbines contributes to 70% of the territorial transformations.

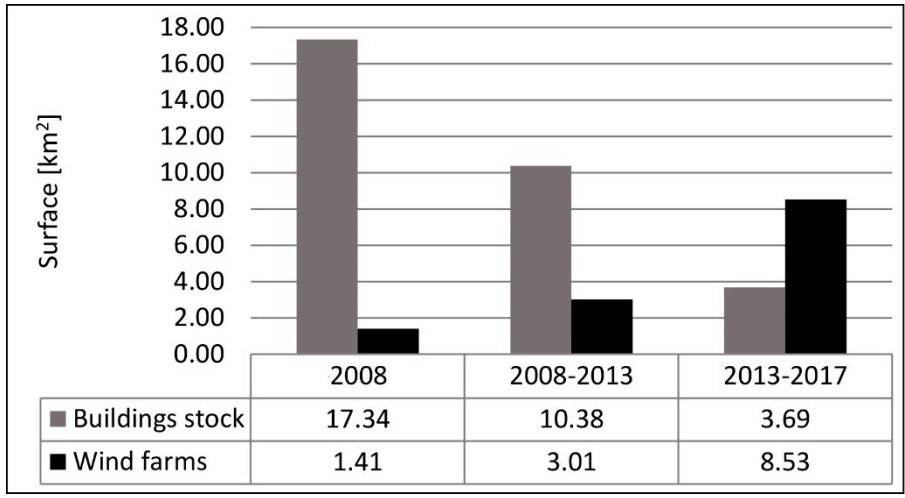

**Figure 10.** Land take deriving from buildings stock and RES farms' growth.

## 5. Conclusions

This work analyzed a peculiar trend in the anthropic use of the territory: to a deceleration in the evolution of the traditional components of the anthropic settlements (residences, industries and infrastructures), there corresponds an increase in the spread of different anthropic elements (the RES farms). During the last decade, Low Carbon Transition policies and $CO_2$ emission targets generated a widespread phenomenon of RES plant settlements in Europe. Regional governments, in Italy, worked towards adopting sectorial plans to regulate the energy sector in accordance with the national and European authorization processes and establish the target of RES power output to be reached in each territory, without strong and effective rules concerning the spatial distribution of the envisaged plants. After the first implementation phase of such complex processes, the Basilicata region, as described in the study area section, is in the first position of the Italian regional ranking for the number of wind turbines installed.

Additionally, the RES plants are not considered as a component of territorial transformation according to current urban planning. Therefore, the spatial distribution of RES plants came about without any formal territorial scheme or provision. This is the demonstration of the current urban

planning weakness (especially in Italy), where planning tools and urban laws are not able to keep up with the fast changes in the anthropization categories (produced by sectorial policies) and therefore, are not suitable to support decision-making processes towards an effective and sustainable development scenario.

In this paper we proposed a quantitative assessment of the impact of RES plants together with building stock development on a regional scale. The assumption is to consider wind turbines as a new component of the territorial settlement whose effects in terms of land take are comparable to traditional settlement categories (residential, industrial, infrastructures, etc.). From the discussion of the results obtained, two main issues emerged:

    a)   The settlement fragmentation degree had been strongly influenced by RES plant settlements in the Basilicata region over the analysis period. Such results represent a critical issue if we consider that the Basilicata region's territorial competitiveness depends on environmental and landscape values, which are the first territorial components affected by an unregulated anthropic fragmentation of the territory.

    b)   The amount of land-take due to RES development between 2008–2017 is about three times the land-take related to urban development (Figure 10).

The weakness of the planning system, in the specific case study of the Basilicata region, lies in the regional planning law (ref. Regional Law 23/99). The absence of a "regional landscape plan" (included in the regional law but still not approved by the Basilicata Region) and of specific regulations on landscape protection and environmental safekeeping, have a strong impact on the governance of these transformations, both on a quantitative and localizing level. In the specific case of wind turbines, this refers to an assessment of territorial suitability that takes into account both specific technological requirements (exposure, altitude, slope, windiness, target to be covered) and compatibility with territorial uses and values suitably acknowledged and documented.

The lack of structured and reliable information on the specific RES plant (location, technical characteristics, operational details etc.) does not allow to monitor the sector from a territorial point of view and neither to define an effective impact assessment procedure to be applied in the ex-ante evaluation of RES projects but supports decision making on the bases of the effectiveness in achieving the policy targets. This is representative of the structural weakness in the instruments and regulations of territorial governance at different scales: if there is no reliable monitoring system for land transformation phenomena, decision makers do not have adequate knowledge to define a territorial development scenario in emerging sectors such as renewable energy.

In this case, the Basilicata region is an explanatory case: it achieved RES production targets defined in the PIEAR (WIN) but it is not able to assess the effects induced by this process of other territorial and landscape components (LOSE).

Concerning the policy implications of this study, if we state that the impact of small and medium-size installations is fully comparable with that of large RES production plants (at least in terms of land take) the future policies/regulations should prefer installations with a higher power output, whereas the small wind turbine plants have to be disadvantaged thorough additional taxations in order to guarantee territorial and landscape integrity of the Basilicata region.

Research development perspectives concern the possibility to improve the monitoring system developed in this work within a wider process of territorial interpretation that documents specific effects on territorial ecosystem components.

**Author Contributions:** All authors contributed equally to this work. In particular, experiment design and writing of the manuscript was developed jointly by all authors. In particular: Data curation, L.S., A.P. and G.P.; Investigation, A.P. and L.S.; Methodology, L.S.; Supervision, B.M.; Validation, F.S. and B.M.; Writing—original draft, A.P.; Writing—review & editing, L.S., F.S. and B.M. All authors have read and agreed to the published version of the manuscript.

**Funding:** This research received no external funding.

**Acknowledgments:** This research has been supported by the Environmental Observatory Foundation of Basilicata Region (FARBAS).

**Conflicts of Interest:** The authors declare no conflict of interest

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
