# Peer review of "Territorial Fragmentation and Renewable Energy Source Plants: Which Relationship?"

_sustainability, doi:10.3390/su12051828_

Round 1
Reviewer 1 Report
Although some results of the manuscript ˝Territorial fragmentation and Renewable Energy Sources: which relationship?˝ could be useful for readers, I cannot recommend this paper for publication in Sustainability. The novelty of the presented findings was not highlighted. The scientific contribution of the paper globally is very poor. The authors used the existing SPX index to assess the fragmentation resulting from territorial transformation due to RES in the Basilicata region. There is no comparison of obtained results with other researches (papers) and thus paper can be interesting to a limited number of the scientist. Also, due to the local character of the research, conclusions can not be generalized. Paper is interesting but more suitable for scientific conferences. Some mistakes in the paper:
1. The description of Figure 1. (in the lines 92-95) doesn't correspond to the Figure.
2. Table 1 is not explained in the text of the manuscript.
3. Figure 4 is not explained in the text of the manuscript.
4. There is a mistake in line 280.
5. Figure 6 is not explained in the text of the manuscript.
6. Some literature is typed with capital letters.
Author Response
REVIEWER 1
Although some results of the manuscript ˝Territorial fragmentation and Renewable Energy Sources: which relationship?˝ could be useful for readers, I cannot recommend this paper for publication in Sustainability. The novelty of the presented findings was not highlighted. The scientific contribution of the paper globally is very poor. The authors used the existing SPX index to assess the fragmentation resulting from territorial transformation due to RES in the Basilicata region. There is no comparison of obtained results with other researches (papers) and thus paper can be interesting to a limited number of the scientist. Also, due to the local character of the research, conclusions can not be generalized. Paper is interesting but more suitable for scientific conferences.
Thank you for your interesting comments. The paper has been restructured and several additions have been made in the introduction, study area, methodology and conclusions. The methodology in particular has been deepened with the addition of new graphs to support it and more explanations. Furthermore an extensive English proofreading was performed.
This research comes from a need linked to the dynamics of territorial transformation that have affected the Basilicata region in recent decades. Characterized by one of the lowest population densities in Italy, the Basilicata region has already been the subject of in-depth study by the authors, as the transformations linked to urban growth do not follow the model of sprawl but that of sprinkling. This has resulted in a very scattered settlement system with all that this implies in terms of territorial fragmentation. What is observed is that in the last ten years, fragmentation has further increased due to the installation of wind turbines. These transformations constitute a new component of the settlement system but are not regulated in any of the urban and/or spatial planning instruments. Although the sprinkling index is not a novelty, the authors consider it a valid tool for monitoring fragmentation linked to the different components of the settlement system. It also makes it possible to consider both the extent and shape of the aggregates and the distance between them.
Since the production of energy from renewable sources is a common and shared need, it is considered useful to test this versatile methodology as it allows to include in the analysis any component of the settlement system. This makes it not only replicable in other areas of study, but also adaptable to different territorial characteristics.
Some mistakes in the paper:
- The description of Figure 1. (in the lines 92-95) doesn't correspond to the Figure.
Thanks, figure 1 has been eliminated because it was considered not useful for the description of the study area.
- Table 1 is not explained in the text of the manuscript.
Table 1 is now explained in the text.
- Figure 4 is not explained in the text of the manuscript.
Thank you. Following the restructuring of the paper, Figure 4 has been removed.
- There is a mistake in line 280.
The error has been resolved.
- Figure 6 is not explained in the text of the manuscript.
The error has been resolved.
- Some literature is typed with capital letters.
The error has been resolved.
Reviewer 2 Report
The paper deals with the use of sprinkling fragmention index to assess regional settlement of RES farms in Basilicata Region (ITALY). The topic of the paper is interesting and relevant with the scope of journal. The paper is adequately structured however it needs a major revision.
These are my comments:
The concept of “territorial fragmentation” is central in the paper. It is better to explain better this idea. The authors give a definition in line 60 but I think it is not enough; 2 line 65-67: the authors affirm that during the last decade “fragmentation” was caused by RES farms. Why? clarify this statement; The fig. 2 is not very clear. It needs more explanation: What is referred to: “the speed of growth changes from 2008-2014”? “In the first section, in fact, area occupied by RES plants grows by only 6% while the built environment grows by 38%”. What is the first section? From the analysis of the fig.2 how do you explain that the RES plant grow of 6%. What is the “built environment”? Line 113: for the same reason of previous comment I don’t understand how the authors obtain RES plants grow of 31%; In the figure 3 The dimension of the legend is too small; The text from line 124 to line 171 should be reduced. It concerns exclusively national (italian) and regional legislation; Line 209: explain better the mean of buffer radius for wind turbines; Line 228: why were the turbines aggregated considering a distance of 250m and buildings of 50m? Line 236: the methodology of Sprinkling index should be further explained. Moreover the reasons for using this index should be highlighted; Fig. 4 is too low resolution; Line 280: check the following: “Error! Reference source not found”.Author Response
The paper deals with the use of sprinkling fragmentation index to assess regional settlement of RES farms in Basilicata Region (ITALY). The topic of the paper is interesting and relevant with the scope of journal. The paper is adequately structured however it needs a major revision.
Thank you for your very interesting comments. The paper has been restructured and several additions have been made in the introduction, study area, methodology and conclusions. The methodology in particular has been deepened with the addition of new graphs to support it and more explanations.
Furthermore an extensive English proofreading was performed.
These are my comments:
- The concept of “territorial fragmentation” is central in the paper. It is better to explain better this idea. The authors give a definition in line 60 but I think it is not enough;
In the introduction the concept of territorial fragmentation has been deepened distinguishing the process of landscape fragmentation from that of territorial fragmentation.
- 2 line 65-67: the authors affirm that during the last decade “fragmentation” was caused by RES farms. Why? clarify this statement;
This research comes from a need linked to the dynamics of territorial transformation that have affected the Basilicata region in recent decades. Characterized by a very scattered settlement system, in the last ten years, fragmentation has further increased due to the installation of wind turbines. These transformations constitute a new component of the settlement system but are not regulated in any of the urban and/or spatial planning instruments.
In the present work, RES installations have been considered as a new component of the territorial settlement. Referring to territorial fragmentation as well as morphological changes of urban areas and their disorganized dispersion in space, RES installation becomes, together with other factors, a cause of territorial fragmentation. It is important to analyse their disposition in the space that in the study area analysed results to have occurred in a fragmented and totally uncontrolled way.
Although the sprinkling index is not a novelty, the authors consider it a valid tool for monitoring fragmentation linked to the different components of the settlement system. It also makes it possible to consider both the extent and shape of the aggregates and the distance between them.
Since the production of energy from renewable sources is a common and shared need, it is considered useful to test this versatile methodology as it allows to include in the analysis any component of the settlement system. This makes it not only replicable in other areas of study, but also adaptable to different territorial characteristics.
In this respect, some additions and changes have been made in the introduction.
- The fig. 2 is not very clear. It needs more explanation: What is referred to: “the speed of growth changes from 2008-2014”? “In the first section, in fact, area occupied by RES plants grows by only 6% while the built environment grows by 38%”. What is the first section? From the analysis of the fig.2 how do you explain that the RES plant grow of 6%. What is the “built environment”?
The section concerning the study area has been reformulated. The graphs showing the comparison between population and components of the settlement system have been eliminated as they are not considered useful for the purposes of the paper.
- Line 113: for the same reason of previous comment I don’t understand how the authors obtain RES plants grow of 31%;
This section has been removed.
- In the figure 3 The dimension of the legend is too small;
Figure 3 now numbered as figure 2 has been revised and integrated with a histogram (in figure 1) that improves its reading.
- The text from line 124 to line 171 should be reduced. It concerns exclusively national (italian) and regional legislation;
The text from line 124 to line 171 has been reduced. We agree that it only concerns national (Italian) and regional legislation but we consider it useful to introduce the context in which the paper is inserted. It is also useful to explain that the need to produce energy from renewable sources does not completely bend the transformation dynamics that have occurred. For the same amount of power produced, in fact, a much smaller number of large scale turbines would have been sufficient.
- Line 209: explain better the mean of buffer radius for wind turbines;
The Buffer radius has been better explained. In addition, a graph representing the relationship between installed power and area occupied by the turbines has been added.
- Line 228: why were the turbines aggregated considering a distance of 250m and buildings of 50m?
The aggregation distance of 250 m for wind turbines is considered the most appropriate to represent the context of the RES territorial component of the Basilicata region. The aggregation of wind turbines allows to consider not only the pitch, the road access and the technical rooms of the plant itself, but also a measure of the total area irreversibly subtracted from its natural vocation.
The aggregation distance of 50 meters for the buildings stock has already been studied in previous works concerning the same territorial context and has been useful to represent a low-density context as it is in the Basilicata region.
In the section 3.2 (Aggregates) some additional explanations have been reported.
- Line 236: the methodology of Sprinkling index should be further explained. Moreover the reasons for using this index should be highlighted;
The methodology used in this work is based on the sprinkling index (SPX), proposed by Romano et al. 2017. As described in the text (the paragraph of the methodology has been integrated), SPX allows to consider the extension and shape of aggregates in addition to their distance from each other. It has been used because it allows to estimate the territorial fragmentation due to different components of the settlement system (in the case study ‘buildings stock’ and ‘wind turbines’). Furthermore, since it has already been used to assess the effects of urban growth in the low density context of the Basilicata region, it is considered useful to make a comparison on the additional contribution from RES installations.
Furthermore a conceptual map has been inserted which summarizes the methodology adopted.
- 4 is too low resolution;
Figure 4 has been removed.
- Line 280: check the following: “Error! Reference source not found”.
Checked, thanks
Reviewer 3 Report
Shortened Review
Topic: Proper embedding of a renewable energies infrastructure into land/seascapes and the related socio-ecological systems is a crucial contemporary challenge. The authors elaborate on this topic which is highly relevant for the audience of the Sustainability journal.
Innovation: The authors apply a GIS based model (SPX assessment) they have presented already in a former paper.
Soundness: The SPX assessment is restricted to a geometrical analysis of features in landscapes. The authors make use of simplistic assumptions on the land coverage of installations and related infrastructure in particular in the case of wind turbines or wind warks. Geometry is the only criterion whilst other crucial values and relationships are left out so far. This results in patters which are very limited and hardly provide evidence for the issues raised in the discussion. The authors are suggested to provide more soundness either with respect to spatial impacts of installations (neighbourhoods, heights, features in the socio-ecological system...) or with respect to detailed examples of impacts of certain policy changes as being discussed in the paper. Figures and tables are difficult to read (font sizes etc) and often suffer from an adequate scale to visualise changes in land coverage and use).
English: The paper needs thorough reworking with respect to the use of the English language in terms of both proper terminology and phrasing.
Some details:
Title: Sources for renewable energy are wind, water, biomass, sunlight etc.; The study considers infrastructure for renewable energy supply (conversion sites). RES (renewable energy conversion sites) constitute a new category of territorial settlement.
Figs 2; 3 change order
Line 131: Make use of EU terminology - do you mean DSO?
Structure needs revision - 2 Study site and 2. Methodology?
Line 258 anthropization - Unknown term - needs definition or to be substituted by suited term.
Fig 4: Legend chafes reader - meaning of different image tiles for low to high fragmentation. Are the tiles displaying different fragmentation geometries falling into the respective fragmentation classes.
Line 286 to 292 Statement is not backed by Fig. 6.
Author Response
Shortened Review
Topic: Proper embedding of a renewable energies infrastructure into land/seascapes and the related socio-ecological systems is a crucial contemporary challenge. The authors elaborate on this topic which is highly relevant for the audience of the Sustainability journal.
Thank you for your very interesting comments. The paper has been restructured and several additions have been made in the introduction, study area, methodology and conclusions. The methodology in particular has been deepened with the addition of new graphs to support it and more explanations.
Furthermore an extensive English proofreading was performed.
Innovation: The authors apply a GIS based model (SPX assessment) they have presented already in a former paper.
The methodology used in this work is based on the sprinkling index (SPX), proposed by Romano et al. 2017. As described in the text (the paragraph of the methodology has been integrated), SPX allows to consider the extension and shape of aggregates in addition to their distance from each other. It has been used because it allows to estimate the territorial fragmentation due to different components of the settlement system (in the case study ‘buildings stock’ and ‘wind turbines’). Furthermore, since it has already been used to assess the effects of urban growth in the low density context of the Basilicata region, it is considered useful to make a comparison on the additional contribution from RES installations.
Soundness: The SPX assessment is restricted to a geometrical analysis of features in landscapes. The authors make use of simplistic assumptions on the land coverage of installations and related infrastructure in particular in the case of wind turbines or wind warks. Geometry is the only criterion whilst other crucial values and relationships are left out so far. This results in patters which are very limited and hardly provide evidence for the issues raised in the discussion. The authors are suggested to provide more soundness either with respect to spatial impacts of installations (neighbourhoods, heights, features in the socio-ecological system...) or with respect to detailed examples of impacts of certain policy changes as being discussed in the paper.
Thanks for the valuable advice. Actually the aim of the paper is not to argue the overall impact of RES installations but to assess the increased fragmentation due to this new component of the settlement system in addition to the contribution of 'buildings stock'. Previous studies had already shown that, despite depopulation and low settlement density, the dynamics of urban growth were leading to a settlement system heavily pulverized, not compatible with the classic dynamics of sprawl. The index thus allows to consider the further contribution of RES technologies in determining a high degree of territorial fragmentation and to compare this result with the previous study.
Figures and tables are difficult to read (font sizes etc) and often suffer from an adequate scale to visualise changes in land coverage and use).
Figures, tables and graphs were replaced and integrated with new elaborations.
English: The paper needs thorough reworking with respect to the use of the English language in terms of both proper terminology and phrasing.
Thanks. The document has been restructured, several additions have been made and an English proofreading was performed
Some details:
Title: Sources for renewable energy are wind, water, biomass, sunlight etc.; The study considers infrastructure for renewable energy supply (conversion sites). RES (renewable energy conversion sites) constitute a new category of territorial settlement.
Thanks for the comment. Actually, by RES farms we mean technologies for the production of energy from renewable sources. Since the theme is territorial fragmentation in the Basilicata region, only the components that contribute to characterize the settlement system as further fragmented have been considered.
Following your advice, the title has been modified.
Figs 2; 3 change order
Thanks, Figure 2 were removed and Figure 3 were replaced.
Line 131: Make use of EU terminology - do you mean DSO?
Thanks for advice. Following the revisions, the structure of the paper has been modified and the part describing the Italian and regional normative has been very synthesized so that those lines have been removed.
Structure needs revision - 2 Study site and 2. Methodology?
Thanks, the structure has been reformulated.
Line 258 anthropization - Unknown term - needs definition or to be substituted by suited term.
We intended the terms “anthropization” as the transformation or adaptation of the environment to meet the needs of humans, or by human activity. In this specific case we refer to all natural or semi-natural surfaces transformed for anthropic activity. These transformations, reversible and irreversible, can indifferently make them impervious or only decrease their permeability.
Fig 4: Legend chafes reader - meaning of different image tiles for low to high fragmentation. Are the tiles displaying different fragmentation geometries falling into the respective fragmentation classes.
Figures, tables and graphs were replaced and integrated with new consideration.
Line 286 to 292 Statement is not backed by Fig. 6.
Figure 6 has been modified to make it more readable. It shows the surfaces falling into each fragmentation class considering only the 'buildings stock' component and both components together ('buildings stock' and 'RES farms'). The aim is to highlight the increase in areas classified as 'high fragmentation' considering the contribution of RES.
Round 2
Reviewer 1 Report
Authors have made significant changes to the manuscript and it is with no doubt improved the paper. Although my opinion is still that scientific contribution of the paper is quite poor, if other reviewers suggest acceptance then my decision is also accept.
Reviewer 2 Report
The authors addressed the suggested comments.
Reviewer 3 Report
The authors managed to improve the quality of their paper with respect to several key domains. The additional work and the elevated level of quality is acknowledged. This particularly applies to both content and design of the visual components of the paper.
Serious weaknesses, however, are still preventing the manuscript from being recommended for publication. There are three fundamental shortcomings
(i) The use of terms is not consistent throughout the paper, e.g., RES plants vs RES farms vs RES installations, whilst the latter is preferable. Several terms are used without proper definition in the text, e.g. anthropic. This is misleading as the common understanding of the term anthropic falls into the realm of philosophy. Another term is densification and there are many more that need to be reconsidered and used precisely. In order to communicate unambiguously, it might be helpful to collaborate with a native English speaking colleague when writing the paper already. The benefit of final proof reading is sometimes restricted to grammatical correctness as in the case of the present paper.
(ii) The paper remains fuzzy with respect to the focus of the study and the research question as well. Core of the study is an analysis of spatio-temporal changes of landscapes and their land use pattern resulting from the introduction of new land use units, namely renewable energy installations and related infrastructure. Readers of the journal are likely to be interested in implications with respect to sustainable regional development. The study presents numbers on the developments in coverage but it remains open whether or not fragmentation is promoting sustainable development, i.e., is a disperse set of installations better than one big park. One can also imagine other independent approaches to assess the coverage, e.g., simple percentages in coverage or in a more sophisticated way fractal dimensions of the distribution of RES or even multifractal spectra indicating certain allocation/aggregation processes.
(iii) The study provides insights into developments in an Italian region. Observations and conclusions are, unfortunately, not sufficiently reflected taking approaches, developments and experiences in other regions into account. In consequence, the impact of the presented study stays within the pilot setting. The global audience of the Sustainability Journal is expecting a broader reflection on the results in order to be enabled to make use of them in a generic manner. The benefit of the study is limited and not fully explored without thorough discussion according to scientific standards.
In summary, the ambition of the study is highly relevand but additional work is needed prior to meeting the standards of an high quality international journal, i.e. Sustainability.